# Cognitive Impairment in Older Cancer Patients Treated with First-Line Chemotherapy

**DOI:** 10.3390/cancers13246171

**Published:** 2021-12-07

**Authors:** Mélanie Dos Santos, Idlir Licaj, Carine Bellera, Laurent Cany, Giulia Binarelli, Pierre Soubeyran, Florence Joly

**Affiliations:** 1Clinical Research Department, Centre François Baclesse, 14000 Caen, France; idlir.licaj@uit.no (I.L.); giulia.binarelli@unicaen.fr (G.B.); f.joly@baclesse.unicancer.fr (F.J.); 2Department of Medical Oncology, Centre François Baclesse, 14000 Caen, France; 3Interdisciplinary Research Unit for the Prevention and Treatment of Cancers (ANTICIPE), National Institute of Health and Medical Research (INSERM), 14000 Caen, France; 4Department of Community Medicine, Faculty of Health Sciences, The UiT Arctic University of Norway, 9010 Tromsø, Norway; 5Clinical and Epidemiological Research Unit, Institut Bergonié, 33076 Bordeaux, France; c.bellera@bordeaux.unicancer.fr; 6Department of Medical Oncology, Clinique Francheville, 24000 Périgueux, France; l.cany@oncoradio24.com; 7Department of Medical Oncology, Institut Bergonie, 33076 Bordeaux, France; P.Soubeyran@bordeaux.unicancer.fr; 8Cancer and Cognition Platform, Ligue Nationale Contre le Cancer, 14000 Caen, France; 9University of Caen Normandie (UNICAEN), Normandie University, 14000 Caen, France

**Keywords:** cancer, chemotherapy, cognitive impairment, elderly, geriatric assessment

## Abstract

**Simple Summary:**

Chemotherapy-related cognitive impairment is frequently reported by patients and can have a negative impact on their quality of life. Elderly patients appear to be particularly at risk for cognitive decline but they are rarely included in studies. Our study investigated cognitive impairment during chemotherapy and its predictive factors among a large elderly population (≥70 years) treated with first-line chemotherapy. The aim was to identify risk factors before starting chemotherapy in order to manage and help elderly patients with decision making.

**Abstract:**

Older cancer patients are vulnerable to chemotherapy-related cognitive impairment. We prospectively evaluated cognitive impairment and its predictive factors during first-line chemotherapy in elderly cancer patients (≥70 years). Cognitive function was evaluated by the Mini-Mental State Examination (MMSE) with adjusted scores for age and sociocultural level. Multidimensional geriatric assessment was performed at baseline and during chemotherapy including the MMSE, Instrumental Activities in Daily Living (IADL), Mini-Nutritional Assessment (MNA), and the Geriatric Depression Scale (GDS15). Quality of life (QoL) was evaluated using the European Organization for Research and Treatment of Cancer (EORTC) QoL Questionnaire (QLQ-C30). Of 364 patients included, 310 had two MMSE evaluations including one at baseline and were assessed. Among these patients, 86 (27.7%) had abnormal MMSE, 195 (62.9%) abnormal MNA, 223 (71.9%) abnormal IADL, and 137 (43.1%) had depressive symptoms at baseline. MMSE impairment during chemotherapy was observed in 58 (18.7%) patients. Abnormal baseline MNA (odds ratio (OR) = 1.87, *p* = 0.021) and MMSE (OR = 2.58, *p* = 0.022) were independent predictive factors of MMSE impairment. These results suggest that pre-existing cognitive impairment and malnutrition are predictive factors for cognitive decline during chemotherapy in elderly cancer patients. Detection and management of these risk factors should be systematically considered in this population before starting chemotherapy.

## 1. Introduction

The incidence of cancer is increasing with an estimated 3.91 million new cases of cancer in Europe in 2018 [1]. Population aging is contributing to the rising number of new cancer cases worldwide and by 2035, older adults are predicted to account for almost 60% of the total incidence of cancer [2].

Chemotherapy-related cognitive impairment (CRCI), also known as “chemobrain”, affects various domains of cognition that can impact daily functioning, quality of life, treatment adherence, and decision making [3]. It affects about 30% of cancer patients and about 70% of cancer patients experience cognitive complaints during or after chemotherapy [4]. This is important for older patients because aging in itself is associated with cognitive modifications and functional decline [5]. Although cognition in cancer patients is an emerging area of research, older adults have not been widely studied. Indeed, very few studies have evaluated cognitive impairment following chemotherapy and always with a very small sample [6,7,8]. A large review of the literature with a total of 2916 elderly treated for a solid or hematological cancer highlighted evidence that more than half of the patients had pre-frailty or frailty with an increased risk of poor tolerance of chemotherapy [9]. Unfortunately, we have poor data on the impact of cancer and associated treatments, in particular in terms of cognitive disorders, in this elderly population that is often underrepresented in clinical trials due to their age [10]. Indeed, the average age of the patients included in the therapeutic trials in oncology is often under 60. Clinical data on cognition are therefore difficult to extrapolate to the elderly, even though they are particularly at risk [7,11]. Furthermore, late-life depression is a common emotional and mental disability in the elderly population that can impact cognitive performance [12]. Therefore, the cognitive effects of chemotherapy in older patients with cancer should not be neglected [8].

A large prospective multicenter study has demonstrated that advanced disease, low MNA score, and poor mobility are predictive factors of early death in elderly cancer patients [13]. However, cognition was not specifically investigated. The Mini-Mental State Examination (MMSE) is a simple screening test for identifying cognitive impairment, which has already been used in the elderly and assessed in this study [14]. In our prospective study, we used the MMSE to assess cognitive impairment during chemotherapy in elderly cancer patients included in the previous one.

## 2. Material and Methods

### 2.1. Participants

Patients aged ≥70 years treated with first-line chemotherapy for various cancers (colon, pancreas, stomach, ovary, bladder, prostate, lung, non-Hodgkin’s lymphoma (NHL), or cancer of unknown primary origin), excluding breast cancer, were prospectively included [13]. Patients with known CNS metastases were excluded. Written informed consent was obtained from all patients included. The protocol was approved by institutional review boards and ethics committees and was conducted in accordance with the Declaration of Helsinki Good Clinical Practices and local ethical and legal requirements (clinical trials: NCT00210249).

### 2.2. Study Design and Measurements

Multidimensional Geriatric Assessment (MGA) was performed by a geriatrician and trained nurse including:-MMSE [15] for cognitive function, with adjusted scores for age and sociocultural level (SCL) according to the French standardization and range of the GRECO work group with abnormal score ≤ 10th percentile of normative values (Appendix A) [16];-Instrumental Activities of Daily Living (IADL) [17], focusing on the ability to perform eight household tasks with abnormal score ≤7;-Geriatric Depression Scale 15-item version (GDS15) [18] for depressive symptoms with abnormal score ≥6;-Mini-Nutritional Assessment (MNA) [19] for nutritional status with abnormal score ≤23.5.

This MGA was performed at baseline before chemotherapy administration, before the second and fourth chemotherapy cycles, and after the sixth cycle or the end of chemotherapy. Supportive care according to usual practices was available according to the needs identified during the MGA. Patients who did not complete a second MMSE evaluation were excluded from the current analysis. Sociodemographic and clinical characteristics were recorded at inclusion: age, sex, marital status, living alone, level of education (pre-primary, primary school certificate, and secondary studies), Eastern Cooperative Oncology Group (ECOG) performance status, cancer site, and stage.

Quality of life (QoL) was evaluated using the European Organization for Research and Treatment of Cancer (EORTC) QoL Questionnaire (QLQ-C30) [20]. A higher score on the functioning or global QoL scale represented a healthier level of functioning or global QoL, and a higher score on the symptom scale represented a worse level of symptomatology.

Chemotherapy treatment was chosen according to standard guidelines at the time of trial registration.

### 2.3. Endpoints

The objective of this analysis was to assess MMSE impairment over time and according to patients’ characteristics and scores for QoL, autonomy, and depression. MMSE impairment was defined as a score ≤ 10th percentile for normal baseline MMSE or loss ≥3 points (10% of total score) for abnormal baseline MMSE [16,21].

### 2.4. Statistical Analysis

Patients’ baseline demographics, clinical characteristics, multidimensional geriatric assessment, and baseline scores for the EORTC QLQ-C30 were analyzed. Further univariate and multivariate logistic regression models were run to compute the odds ratios (ORs) of impaired MMSE and their 95% confidence intervals (CIs). The regression models were built according to a validated approach [22]. We performed univariate regressions for each covariate and included those significant at the 20% level in the multivariate model (the full model). Covariates no longer significant in the full model using Wald statistics were excluded. Log likelihood tests were performed to compare the goodness of fit between the reduced model and the full model. The list of covariates included: age (<80; ≥80 years), sex (male; female), ECOG performance status (0–1; 2; 3–4), living alone (yes; no), tumor site (hematologic; digestive; lung; uro gynecologic), disease extension (localized/ IPI 0–1; metastatic/IPI 2–3), MMSE at baseline (normal; abnormal), GDS15 (<6; ≥6), IADL (<7; ≥7), and MNA (<23.5; ≥23.5). The domain-specific EORTC QLQ-C30 was considered as quantitative continuous variables divided into quintiles (i.e., each domain had 5 categories of 20 points including a clinical change set at 10–20 points). Statistical significance was set at *p* < 0.05 and analysis was performed with STATA version 15 (Stata Corp, College Station, TX, USA).

## 3. Results

Between September 2002 and September 2005, 364 patients from 12 centers in Southwest France were included in this trial. Patients were recruited in two cancer referral centers and 10 community hospitals in their Departments of Oncology and Gerontology. Twelve patients had not completed the baseline MMSE, and 42 patients did not have a second assessment over time, which left 310 eligible and evaluable patients (Figure 1). Baseline demographic and clinical characteristics are presented in Table 1. Median age was 77.4 (range: 70–93) with 31% aged over 80, 59% of participants were men and 64.7% were married. Tumor site was mainly digestive (42.9%) and NHL (31.3%) with mostly metastatic disease for solid tumors (62.3%) and localized for NHL (57.7%). Almost half of the patients received standard treatment.

### 3.1. Baseline MGA

The median MMSE score was 26.2 (range: 9 to 30). According to age and the SCL cut-off point, 86 (27.7%) patients had abnormal MMSE baseline scores (Table 2). Two hundred and twenty-three (71.9%) patients had an abnormal IADL score and 137 (43.1%) had depressive symptoms. Only 111 (35.8%) patients had a good nutritional status (MNA > 23.5).

### 3.2. Baseline QoL Scores

Baseline QoL scores are summarized in Table 3. The poorest functioning was reported for global QoL (57.2 ± 20.2) and role function (69.4 ± 33.0). The most severe symptom was fatigue (38.3 ± 28.8). However, these scores were similar to those of the EORTC reference values for all cancer patients ≥70 years [23].

### 3.3. MMSE Impairment

MMSE impairment was observed for 58 (18.7%) patients during chemotherapy, including 10 patients with an abnormal initial MMSE. Among these patients, the median loss was 3 points (range: 1–16). The median time until MMSE impairment was 15.4 (5.3–26.4) months. Potential factors associated with MMSE impairment were assessed at baseline in univariate analysis (Table 4). Abnormal baseline MNA (*p* = 0.031) and MMSE scores (*p* = 0.048), living alone (*p* = 0.029), and several QLQ-C 30 items on both functional and symptom scales were significantly associated with MMSE impairment during chemotherapy.

All these significant variables as well as those clinically relevant (age and sex) were included in the multivariate model (Table 4). Abnormal MMSE (OR = 2.58, *p* = 0.022) and MNA scores (OR = 2.58, *p* = 0.021) at baseline were independent factors associated with risk of MMSE impairment. Concerning the EORTC QLQ-C30, three functioning scales were predictive of MMSE impairment: physical function (*p* = 0.048), emotional function (*p* = 0.020) and social function (*p* = 0.021). Pain was also independently associated with MMSE impairment (*p* = 0.003).

## 4. Discussion

This large multicenter cohort with elderly cancer patients (≥70 years) treated with first-line chemotherapy investigated MMSE impairment during chemotherapy and its predictive factors. Very few studies have evaluated cognitive impairment under chemotherapy in older patients and always with a very small sample. To our knowledge, the present study is the largest in the literature [6,24,25,26,27]. Among the 310 patients evaluated, 58 (18.7%) patients experienced MMSE impairment. Malnutrition (MNA) and cognitive impairment (MMSE) at baseline were significantly associated with cognitive impairment during chemotherapy. Pain was also a significant predictive factor of MMSE impairment.

CRCI is a common symptom during chemotherapy and may influence adherence to treatments, impair quality of life, and lead to long-term cognitive impairments [28]. In our patients, pre-existing cognitive impairment appeared to be a risk factor for cognitive decline during chemotherapy as compared to normal baseline MMSE, with more than two-fold higher odds of MMSE impairment. Cognitive impairment was identified at baseline in 86 (27.7%) patients with an abnormal MMSE score according to age and SCL cut-off point [16]. The original validation study for the MMSE suggested a cut-off score < 24 indicating cognitive impairment [15]. However, this may not be adapted for older adults, particularly for those with less education. Indeed, several studies have shown that cognitive performance is related to age and educational level, with some studies using the MMSE score [29,30,31,32].

We also found that an abnormal MNA at baseline was associated with cognitive decline during chemotherapy with almost two-fold higher odds of MMSE impairment as compared to normal MNA. The pro-inflammatory effects of some forms of nutrition are thought to be linked to cognitive function [33]. Chemotherapy can also impact nutritional status and thus worsen the MNA score during treatment [34]. Nutritional status has become a domain of interest in oncology. It has been associated in several studies with systemic treatment-related outcomes and also mortality, notably in older cancer patients [13,35,36]. Indeed, recent data suggest that lifestyle factors such as nutrition and physical activity have an impact on cancer outcomes and on CRCI [37,38].

A prospective study in 202 older cancer patients (≥70 years) investigated the predictive value of the MGA before chemotherapy [36]. Low MMSE and MNA scores at baseline appeared independently related to the probability of not completing chemotherapy. Twenty-one (10%) of the 202 patients had an abnormal MMSE at baseline (≤24), and statistically significant MMSE impairment was evidenced in the 51 patients who underwent post-chemotherapy evaluation. Other prospective studies have shown that pre-existing cognitive impairment may also be a risk factor for chemotherapy toxicity [39,40].

Concerning EORTC QLQ-C 30 patient-reported outcomes, there was a significant relationship between pain score at baseline and MMSE impairment during chemotherapy. Some research suggests that pain-related negative emotions and stress potentially impact cognitive functioning [41,42]. However, the use of analgesics, especially opioids, can impact cognitive performance, including the MMSE score [43,44]. Unfortunately, we could not evaluate the impact of pain medication since information on concomitant medication was not collected. Low scores on emotional, social, and physical functioning were also associated with MMSE impairment. Although the MMSE is the most commonly used cognitive screening test before oncological treatment, a recent prospective study suggests that the Montreal Cognitive Assessment (MoCA) [45] is the most relevant to screen cognitive impairment in older patients with cancer [46]. It would be interesting to see whether the MoCA confirms the MMSE findings in our patients.

We found that almost half of the patients presented depressive symptoms before the start of chemotherapy according to the GDS15 assessment. Indeed, depression is a very common symptom in cancer patients, especially in the elderly population [47]. However, we did not find a significant association between depression and cognitive impairment. This can be explained by the fact that psychological factors such as depression, anxiety, or fatigue seem to impact more clearly cognitive complaints than cognitive performance [48]. Indeed, imaging studies suggest a compensatory activation of additional brain regions to maintain the performance on neuropsychological tests of these patients with anxio-depressive symptoms [49].

This study has some strengths. First, our population includes only elderly patients with a median age over 77 and an age range between 70 and 93 years old, which is extremely rare in oncology studies. While cognitive impairment is an emerging area of research in oncology, elderly patients have so far received little attention. Second, our population seems representative of the general population with QoL scores comparable to those of the EORTC reference values for all cancer patients ≥70 years [23]. The study also has some limitations. For example, we were unable to include breast cancer patients, in whom the chemobrain has been widely studied. The data available on the chemotherapy protocols that the patients had followed was only partial. In addition, 42 (11%) of the patients included in the study could not be evaluated because they have not completed a second MMSE assessment. The sample of the study analyzed remains large, but it should be emphasized. Moreover, it should be noted that the MMSE is a screening tool for cognitive impairment. It would be interesting to confirm these results with the use of specific tests evaluating the cognitive domains most affected by cancer treatments such as the Hopkins Verbal Learning Test-Revised (HVLT-R) for learning performance and episodic memory and the Trail Making Test (TMT) for executive functions and processing speed [11]. Moreover, it would have been interesting to have a control age-matched cohort to support our results in order to exclude an impact of normal aging. Some cognitive decline was observed early during chemotherapy (from 5 months). This observation seems more in favor of an impact of the treatment than of normal aging. Finally, the fact that the data were collected between 2002 to 2005 may raise questions of the current validity and applicability of our results. However, the study population mainly presented colon cancer or non-Hodgkin lymphoma; tumors for which the standard treatments are currently significantly the same. Nonetheless, this study has the merit of providing data in elderly patients, who are unfortunately very poorly represented in clinical studies, and makes it possible to identify predictive factors of cognitive decline during chemotherapy. Although nutritional status and cognition have recently become areas of focus in oncology, there is still a lot of prevention that needs to be carried out in these areas and our results remain relevant and are still a topical issue.

## 5. Conclusions

Older patients are more vulnerable to chemotherapy toxicity and notably to CRCI [6]. This is problematic since the number of older patients with cancer is increasing, as is the number of elderly cancer patients with long-term survival [2]. Our findings suggest that pre-existing cognitive impairment and malnutrition are independent predictive factors of cognitive decline during chemotherapy in elderly cancer patients. The detection and management of these risk factors with a systematic MGA should be mandatory in this population before they begin chemotherapy in order to help with decision making. Pain symptoms should also be carefully addressed.

## Figures and Tables

**Figure 1 cancers-13-06171-f001:**
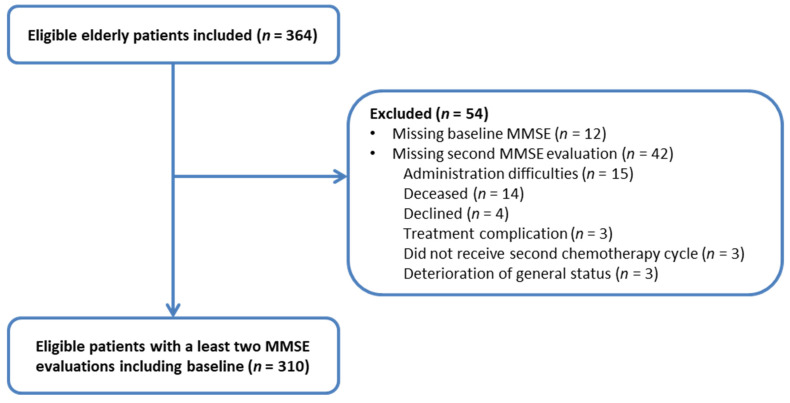
Patient inclusions in study investigating cognitive impairment in older cancer patients receiving first-line chemotherapy. Abbreviations: MMSE, Mini-Mental State Examination.

**Table 1 cancers-13-06171-t001:** Baseline demographic and clinical characteristics of elderly patients receiving first-line chemotherapy (*n* = 310).

Demographic and Clinical Characteristics	No.	%
**Median age, years (range)**	77.4 (70–93)	
**≥80**	96	31.0
**Sex**		
**Male**	183	59.0
**Female**	127	41.0
**ECOG performance status**		
**0–1**	234	75.5
**2**	52	16.7
**3–4**	17	5.5
**Missing data**	7	2.3
**Living alone**	84	27.3
**Marital status**		
**Single**	12	3.9
**Married**	200	64.5
**Widower**	79	25.5
**Divorced**	18	5.8
**Missing data**	1	0.3
**Education level**		
**Pre-primary**	64	20.6
**Primary school certificate**	151	48.7
**Secondary studies**	61	19.7
**Higher studies**	33	10.7
**Missing data**	1	0.3
**Tumor site**		
**NHL**	97	31.3
**Colon**	83	26.8
**Stomach**	32	10.3
**Lung**	31	10.0
**Pancreas**	18	5.8
**Prostate**	16	5.2
**Bladder**	15	4.8
**Ovary**	14	4.5
**Primary unknown**	4	1.3
**Disease extension**		
**Solid tumors**	213	68.7
**Localized**	74	23.9
**Metastatic**	139	44.8

**NHL**	97	31.3
**aaIPI score 0–1**	56	18.1
**aaIPI score 2–3**	41	13.2

Abbreviations: ECOG, Eastern Cooperative Oncology Group; NHL, non-Hodgkin’s lymphoma; aaIPI, age-adapted International Prognostic Index.

**Table 2 cancers-13-06171-t002:** Multidimensional geriatric assessment data for all eligible patients (*n* = 310).

Baseline Geriatric Evaluation	Normal Score		Abnormal Score *		Missing		Overall	
	No.	%	No.	%	No.	%	Median	Range
MMSE	224	72.3	86	27.7	0	0	27.5	9 to 30
GDS 15	167	53.9	137	44.2	6	1.9	5	0 to 13
IADL	86	27.8	223	71.9	1	0.3	6	0 to 8
MNA	111	35.8	195	62.9	4	1.3	21.5	6 to 28

Abbreviations: MMSE, Mini-Mental Status Examination; GDS 15, Geriatric Depression Scale; IADL, Instrumental Activities in Daily Living; MNA, Mini Nutritional Assessment. * Abnormal scores defined as: MMSE: age and education-adjusted cut-off point; GDS15 ≥ 6; IADL ≤ 7; MNA ≤ 23.5.

**Table 3 cancers-13-06171-t003:** Baseline scores for the EORTC QLQ-C30 (*n* = 310) and reference values.

EORTC QLQ-C30	Study Sample			Reference Values *	
	No.	Mean	No.	Mean	No.
Global health status/QoL	304	57.2	20.2	60.6	25.1
Physical functioning	305	74.8	22.7	72.1	25.4
Role functioning	304	69.4	33	70.7	34.1
Emotional functioning	305	73.5	22.8	76.1	23.2
Cognitive functioning	305	77.6	23.6	81	22.4
Social functioning	303	81.6	26.7	78.2	28.2
Fatigue	305	38.3	28.8	35.7	29
Nausea and vomiting	305	7.8	18	9.1	19.2
Pain	305	25.6	31.7	25.9	30.5
Dyspnoea	305	24.8	30.3	23.1	29.6
Insomnia	303	30.2	3.7	26.4	31.3
Appetite loss	304	34	37.5	22.4	33.2
Constipation	302	22.7	33.1	21.7	31.2
Diarrhoea	302	13.2	26.9	8.9	20.7
Financial difficulties	303	2.4	10.6	8.5	20.6

* EORTC references values for all cancer patients ≥70 years. From Scott NW et al.: EORTC QLQ-C30 Reference Values Manual 2008. Abbreviations: EORTC QLQ-C30, European Organization for Research and Treatment of Cancer Quality of Life. Questionnaire.

**Table 4 cancers-13-06171-t004:** Factors associated with MMSE impairment during chemotherapy (*n* = 310).

Factor	Univariate			Multivariate		
	OR	95% CI	*p*	OR.	95% CI	*p*
**Age (years)**			0.353			0.372
<80	1 (reference)					
≥80	0.74	0.38–1.40		0.72	0.35–1.48	
**Sexe**			0.603			0.632
Male	1 (reference)					
Female	1.17	0.65–2.10		1.18	0.60–2.32	
**ECOG performance status**				Excluded		
0–1	1 (reference)					
2	1.45	0.70–3.02	0.314			
3–4	2.02	0.67–6.06	0.210			
**Living alone**			**0.029**			0.097
No	1 (reference)					
Yes	**2.33**	**1.09–4.99**		2.02	0.88–4.65	
**Tumor site**				Excluded		
Hematologic	1 (reference)					
Digestive	0.92	0.46–1.81	0.805			
Lung	1.05	0.38–2.95	0.921			
Uro Gynecologic	1.25	0.52–2.99	0.611			
**Disease extension**			0.494	Excluded		
Localized/IPI 0–1	1 (reference)					
Metastatic/IPI 2–3	1.23	0.68–2.21				
**MMSE at baseline** *			**0.048**			**0.022**
Normal	1 (reference)			1 (reference)		
Abnormal	2.07	**1.09–4.31**		**2.58**	**1.14–5.84**	
**GDS15**			0.089	Excluded		
<6	1 (reference)					
≥6	1.66	0.93–2.98				
**IADL**			0.099	Excluded		
>7	1 (reference)					
≤7	1.82	0.89–3.71				
**MNA**			**0.031**			**0.021**
>23.5	1 (reference)					
≤23.5	**1.91**	**1.21–3.24**		**1.87**	**1.18–3.58**	
**EORTC QLQ-C30**						
Global health status/QoL	**0.8**	**0.65–0.97**	**0.029**	0.81	0.65–1.00	0.054
Physical functioning	**0.77**	**0.62–0.95**	**0.016**	**0.77**	**0.59–0.98**	**0.048**
Role functioning	**0.79**	**0.65–0.96**	**0.017**	0.82	0.66–1.03	0.087
Emotional functioning	**0.81**	**0.67–0.98**	**0.034**	**0.82**	**0.68–0.93**	**0.02**
Cognitive functioning	0.87	0.73–1.05	0.162	Excluded		
Social functioning	**0.77**	**0.64–0.92**	**0.004**	**0.8**	**0.67–0.97**	**0.021**
Fatigue	1.21	0.98–1.50	0.075	1.17	0.92–1.47	0.198
Nausea and vomiting	0.8	0.39–1.65	0.548	Excluded		
Pain	**1.81**	**1.27–2.59**	**0.001**	**1.77**	**1.21–2.59**	**0.003**
Dyspnoea	0.99	0.74–1.32	0.933	Excluded		
Insomnia	1.28	0.92–1.80	0.146	Excluded		
Appetite loss	1.25	0.90–1.75	0.183	Excluded		
Constipation	1.28	0.71–2.29	0.412	Excluded		
Diarrhoea	0.87	0.44–1.73	0.695	Excluded		
Financial difficulties	0.87	0.73–1.05	0.164	Excluded		

Abbreviations: IPI, International Prognostic Index; MMSE, Mini-Mental Status Examination; GDS 15, Geriatric Depression Scale; IADL, Instrumental Activities in Daily Living; MNA, Mini Nutritional Assessment; EORTC QLQ-C30, European Organization for Research and Treatment of Cancer Quality of Life Questionnaire. * Abnormal scores defined with age and education-adjusted cut-off point.

## Data Availability

The data presented in this study are available on request from the corresponding author. The data are not publicly available accordingly to good clinical practice (GCP) guidelines.

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
