# Peer review of "Cognitive Impairment in Older Cancer Patients Treated with First-Line Chemotherapy"

_cancers, 2021, doi:10.3390/cancers13246171_

Round 1

Reviewer 1 Report

I thank the authors for implementing the article following my remarks. In this form the article can be published and does not require further changes.

Author Response

We would like to thank the reviewer for his fruitful expertise and comments.

Dr Mélanie Dos Santos

Reviewer 2 Report

We thank the authors for addressing the questions raised by the reviewers. However, their answers to these questions must be incorporated into the manuscript. The reason these questions have been raised is because we, the reviewers believe that the scientific value of answers to these questions are missing from the manuscript. Replying to the reviewers serves little purpose if they are not incorporated into the manuscript. If the authors are unable to provide data to support the requests made by the reviewers, then this information should be listed as a limitation of the current study or included as potential future directions.

The numbers in table 1 do not add up:

Sex = 360

ECOG performance = 303 (but = 100%)?

Most of the other categories add up to 309 (not 310) – if data is missing, then please indicate the number of samples for which data is missing.

It is unclear why the authors have highlighted large sections of the manuscript since the text in these highlighted sections (other than two paragraphs in red – one in the intro and one in the discussion) are identical to the original text.

Regarding novelty, the authors have stated "this is the first study to explore cognitive impairment under chemotherapy in older patients with cancer". However, a simple Google search reveals articles dating back to at least 2010 that have reported investigations relating to CRCI in older patients. These studies include the use of non-treated as well as healthy controls and the use of more robust tests of cognition. The authors are referred to the following article and references within: Magnuson, Mohile and Janelsins, (2016) Curr Geriatr Rep. 5(3): 213–219. doi: 10.1007/s13670-016-0182-9

Also, please see:

Lange M, et al. (2014) Eur J Cancer. 50:2181–9.

Mandelblatt JS, et al. (2014) J Clin Oncol. 32:1909–18.

Ahles TA, et al. (2010) J Clin Oncol. 28:4434–40.

Pergolotti et al. (2020) J Geriatr Oncol. 11(2): 237–243.

Based on these findings, the novelty and significance of this manuscript is substantially lowered.

Author Response

We would like to thank the reviewer for his fruitful expertise and comments.

Round 2

Reviewer 2 Report

We thanks the authors for addressing the reviewers comments and for incorporating suggestions into their manuscript.

Minor comment:

Please remove the sentence "Nevertheless, the patients were their own controls since all MMSE postbaseline measures were compared to the baseline of each patient limiting this bias".

This statement is incorrect since the authors are comparing global cognition before and after treatment and as such, subjects cannot be considered their own controls or else we will not require placebo/control groups in clinical trials and/or experiments.

The authors should include a statement in the introduction indicating that "Very few have evaluated cognitive impairment following chemotherapy and always with a very small sample (Ahles TA, et al. (2010); Lange M, et al. (2014); Mandelblatt JS, et al. (2014)). The Ahles study is important as it contained an age-matched control group.

Some language editing is required in particular with the use of tenses.

Author Response

Thank you

This manuscript is a resubmission of an earlier submission. The following is a list of the peer review reports and author responses from that submission.

Round 1

Reviewer 1 Report

The article is interesting and addresses the problem of cognitive decline in cancer patients undergoing chemotherapy.

The work is well developed although in my opinion it needs some changes.

The introduction completely misses a mention to the importance of late life depression in the development of cognitive disorders. In this regard, I recommend to see the article: Cieri F, Esposito R, Cera N, Pieramico V, Tartaro A, by Giannantonio M. Late-Life Depression: Modifications of Brain Resting State Activity. J Geriatr Psychiatry Neurol. 2017 May; 30 (3): 140-150. doi: 10.1177 / 0891988717700509. Epub 2017 Mar 30. PMID: 28355945.

It seems important to introduce this point given that 137 patients have alterations to the GDS.

Furthermore, did the authors investigate whether the type of tumor affect cognitive function? Are there any data in the literature on this matter? I refer for example to a comparison between lymphomas with systemic diffusion compared with tumors more localized in one organ.

Moreover the authors should underline the importance of cognitive reserve closely linked to education levels. What role does it play in cognitive impairment? is it possible to hypothesize a support in delaying the onset of cognitive decline in cancer patients? What role does the level of education play? Was this element taken into account as covariate in the statistical analysis to highlight a possible influence on the results?

In my opinion, the discussion is too short. The same points of the introduction should be discussed (role of depression, influence of education levels, variations according to tumor type).

Furthermore, the study limits section needs to be expanded: although the study field is significant, the work has important limitations that should be highlighted. The first of all is the lack of neuropsychological tests that explore the different cognitive functions (prose memory test (Babcock story) to evaluate prose memory, Frontal Assessment Battery (FAB) to screen for global executive functions, Trail Making Test etc. etc.). The MMSE alone appears to be limiting.

Furthermore, in the inclusion criteria no reference is made to any therapies other than chemotherapy (psychoactive drugs? pain drugs?) which generally affect cognitive functions.

Reviewer 2 Report

The manuscript by dos Santos and colleagues investigates chemotherapy associated cognitive impairment in elderly subjects over the age of 70. The authors report that 27.7% of individuals receiving chemotherapy showed signs of cognitive impairment.

Comments

  • A significant limitation of this investigation is the absence of a ‘control’, normal, age-matched cohort. One cannot determine whether the cognitive decline observed in these elderly patients is due to normal ageing. It is acknowledged that the authors have provided normative values adjusted for age and SCL, however, could other variables investigated in this cohort be attributable to the ‘normal’ age-matched population from the same study population? Also, did the present study employ the GRECO version of the “Mini Mental State”
  • In the introduction the authors indicate that older adults are predicted to account for almost 60% of the total incidence of cancer by 2035, fourteen years from now. The current investigation reports on data that was collected between 2002 and 2005. How representative is this data of current elderly chemotherapy patients (16 years later)? Cohorts currently over the age of 70 receiving chemotherapy could speculatively be more educated, be better nourished, receive better medical treatment as well as more advanced and personalized chemotherapeutic treatments. These points raise questions pertaining to the validity and applicability of the results and conclusions of the present report.
  • The authors report that 27.7% of subjects had abnormal MMSE scores at baseline. During treatment, 58 patients (18.7%) experienced MMSE impairment, including 10 patients with abnormal MMSE scores at baseline. The median loss was 3 points over a median period of 15.4 (5.3-26.4) months in this elderly population. Could this have been due to the ‘normal’ ageing process? Also, does this mean that MMSE scores of the other 76 individuals who had abnormal MMSE scores at baseline improved following treatment? If yes, then this must be reported. This is a substantial, potentially significant finding.
  • The methods indicate that MGA was performed at baseline, before the second and fourth chemotherapy cycles, and after the sixth cycle or the end of chemotherapy. Does this mean that MMSE tests (as well as others) were performed 3 times on some participants? Also, if the MGA was performed 3X, were the participants advised on nutritional (as well as other) requirements at baseline and midway through treatment?
  • The formatting of Table 3 and Table 4 must be corrected.
  • Please describe/explain the SCL numbers (SCL1, 2, 3 and 4) presented in the Annex.

Reviewer 3 Report

  1. Patients who did not complete a second MMSE evaluation were important to assess the data in the manuscript. That should be addressed in Discussion.
  2. The authors indicate strengths of this study. However, there were several limitations. Primary tumor sites and medications of the treatment of cancers should affect the results of this study. These should be indicated as limitations.
  3. Table 3 and 4 do not have a quality to be published in our Journal. The reviewer may misunderstand the meaning of the Tables.
